# CSA Antisense Targeting Enhances Anticancer Drug Sensitivity in Breast Cancer Cells, including the Triple-Negative Subtype

**DOI:** 10.3390/cancers14071687

**Published:** 2022-03-26

**Authors:** Silvia Filippi, Elena Paccosi, Alessio Balzerano, Margherita Ferretti, Giulia Poli, Juri Taborri, Stefano Brancorsini, Luca Proietti-De-Santis

**Affiliations:** 1Laboratory of Molecular Genetics of Aging, Department of Ecology and Biology, University of Tuscia, 01100 Viterbo, Italy; silvia.filippi@unitus.it (S.F.); e.paccosi@unitus.it (E.P.); alessio.balzerano@unitus.it (A.B.); margheferre96@gmail.com (M.F.); 2Unit of Molecular Pathology, Department of Experimental Medicine, Section of Terni, University of Perugia, 06100 Perugia, Italy; poligiulia.mail@gmail.com (G.P.); stefano.brancorsini@unipg.it (S.B.); 3Department of Economics, Engineering, Society and Business Organization, University of Tuscia, 01100 Viterbo, Italy; juri.taborri@unitus.it

**Keywords:** antisense oligonucleotide (ASO), proliferation, breast cancer, triple-negative subtype, drug sensitivity

## Abstract

**Simple Summary:**

Breast cancer (BC), the most frequent malignancy in woman, shows a high rate of cancer recurrence and resistance to treatment, particularly in Triple-Negative Breast Cancer (TNBC) subtype. Starting from the observation that different subtypes of BC cells, including the TNBC one, display an increased expression of Cockayne Syndrome group A (CSA) protein, which is involved in multiple functions such as DNA repair, transcription and in conferring cell robustness when it is up-regulated, we demonstrated that CSA ablation by AntiSense Oligonucleotides (ASOs) drastically impairs tumorigenicity of BC cells by hampering their survival and proliferative capabilities without affecting normal breast cells. Suppression of CSA does result in lowering the IC_50_ value of Oxaliplatin and Paclitaxel, two commonly used chemotherapeutic agents in breast cancer treatment, allowing the use of a very low dose of chemotherapeutic that is non-toxic to the normal breast cell line. Finally, CSA ablation restores drug sensitivity in oxaliplatin-resistant cells. Based on these findings, we can conclude that CSA may be a very attractive target for the development of new specific anticancer therapies.

**Abstract:**

Breast cancer (BC) is the most common cancer with the highest frequency of death among women. BC is highly heterogenic at the genetic, biological, and clinical level. Despite the significant improvements in diagnosis and treatments of BC, the high rate of cancer recurrence and resistance to treatment remains a major challenge in clinical practice. This issue is particularly relevant in Triple-Negative Breast Cancer (TNBC) subtype, for which chemotherapy remains the main standard therapeutic approach. Here, we observed that BC cells, belonging to different subtypes, including the TNBC, display an increased expression of Cockayne Syndrome group A (CSA) protein, which is involved in multiple functions such as DNA repair, transcription, mitochondrial homeostasis, and cell division and that recently was found to confer cell robustness when it is up-regulated. We demonstrated that CSA ablation by AntiSense Oligonucleotides (ASOs) drastically impairs tumorigenicity of BC cells by hampering their survival and proliferative capabilities without damaging normal cells. Moreover, suppression of CSA dramatically sensitizes BC cells to platinum and taxane derivatives, which are commonly used for BC first-line therapy, even at very low doses not harmful to normal cells. Finally, CSA ablation restores drug sensitivity in oxaliplatin-resistant cells. Based on these results, we conclude that CSA might be a very attractive target for the development of more effective anticancer therapies.

## 1. Introduction

BC is the most frequent malignancy in woman globally impacting around 2–2.5 millions of individuals each year. BC is a heterogeneous disease with molecular subtypes characterized by biological distinctness and different behavior [1]. Currently, clinical practice classifies BC in four subtypes, based on the molecular expression of Estrogen Receptor (ER), Progesterone Receptor (PR), Human Epidermal growth factor Receptor 2 (HER2) and Ki67 proliferative marker: (i) Luminal A (ER+/PR+/HER2-/lowKi-67), (ii) Luminal B (ER+/PR+/HER2-/+/high Ki-67), (iii) HER2-overexpression (ER-/PR-/HER2+), and (iv) Triple Negative Breast Cancers/TNBCs (ER-/PR-/HER2-) [2].

Cytotoxic chemotherapy remains the mainstay for neo-adjuvant treatment and for the first phase of the adjuvant one, although, in the receptor-positive subtypes, it can be combined or even replaced during the oncological therapy, with hormonal and/or biological treatments [3]. Moreover, despite the promise of new targets and biological agents, such as Poly (ADP-Ribose) Polymerase (PARP) inhibitors [4] and atezolizumab for patients with germline BRCA mutations and for PD-L1 positive population [5], respectively, to date, chemotherapy represents the main therapeutic option for the metastatic treatment of the TNBC.

In addition to Adriamycin, platinum and taxane derivatives are the major chemotherapy drugs used in BC therapy. The former induce both intra- and inter-strand DNA crosslinks, leading to a cell death mainly due to the interference with transcription or DNA replication processes [6]. The latter, instead, affect microtubules depolymerization, leading to the stabilization of both cytoskeleton and microtubules structures, such as the metaphase plate and the intercellular bridge; this event induces cell death through mitotic catastrophe or cytokinesis failure [7]. While the above described ones are first line chemotherapy drugs that showed to be very effective in BC treatment regardless of their genetic background, unfortunately two main factors are responsible of their long-term failure. Their first limiting factor is the scarce selectiveness between highly proliferating normal cells (epithelia and hematopoietic tissues) and cancer cells. This non selectivity gives rise to severe collateral effects that limit the dose regiment. The second limiting factor is that, despite these drugs show to be largely effective at the beginning of BC treatment, cancerous cells often develop, by different mechanisms such as an increased repair of drug-induced DNA damage and evasion of apoptosis, an acquired resistance which is responsible for tumors relapse [8,9,10].

Therefore, the identification of both new molecular targets and new biological therapies is required to further improve treatment strategies with the current list of chemotherapy options.

*CSA* gene, localized on chromosome 5q12-q13, codes a 44 kDa protein of 396 amino acids that belongs to the family of WD-40 (W = tryptophan, D = aspartic acid) repeat proteins, known for coordinating interactions among multi-protein complexes [11]. CSA protein is a component of an ubiquitin E3 ligase complex containing Cullin 4 (CUL4), Ring Box-1 (RBX1), and Damage Specific DNA Binding Protein 1 (DDB1) [12,13]. It was first characterized as a DNA repair protein, playing a role in the Transcription Coupled Repair (TCR), a sub-pathway of Nucleotide Excision Repair (NER) devoted to the rapid removal of the transcription-blocking lesions located on the coding strands [14,15,16,17,18]. However, more recently, several findings indicated that CSA is a multifaceted protein implicated not only in DNA repair, but also in the modulation of gene expression and in the triggering of cell division [19,20]. Indeed, CSA, in concert with CSB, participates in the ubiquitination and degradation of tumor suppressor p53 as well as of Activating Transcription Factor 3 (ATF3), thus counteracting the pro-apoptotic responses induced upon cellular stresses, such as DNA damage and Endoplasmic Reticulum (ER), hypoxia, and oxidative stresses [18,21,22,23,24]. Furthermore, Paccosi and collaborators demonstrated that Protein Regulator of Cytokinesis 1 (PRC1) is ubiquitinated and degraded by CSA and CSB proteins during cytokinesis [19].

In this paper we showed that CSA protein is up-regulated in BC cell lines and that its ablation *per se* reduces the tumorigenicity of cancer cells by tipping the balance towards cell cycle arrest and death and away from cell proliferation and survival. Furthermore, we showed that CSA suppression makes BC cells more sensitive to either platinum derivatives or taxanes treatments.

## 2. Materials and Methods

### 2.1. Cell Lines

One normal (MCF10A) and three breast cancer (MCF-7, MDA-MB231, and T47D) cell lines were used. MCF-10A is a non-tumorigenic breast cell line and it was grown in DMEM/F12 (Thermo Fisher, Waltham, MA, USA) medium supplemented with 20 ng/mL Human Epidermal Growth Factor (hEGF) and 10 ng/mL bovine insulin. MCF7 were cultured in Dulbecco’s Modified Eagle Medium (DMEM, Invitrogen, Waltham, MA, USA) with 10 ng/mL of bovine insulin. T47D and MDA-MB-231 were grown in RPMI-1640 medium. All culture mediums were supplemented with 10% Foetal Bovine Serum (FBS, Invitrogen), 2 mM Glutamine (Lonza), and 40 µg/mL of Gentamicin (Sigma-Aldrich: ST.Louis, MO, USA). MDA-MB-231^resistant^ cell line was obtained according to McDermott et al., 2014 [25]. MCF-10A, MCF-7, MDA-MB231, and T47D were purchased from American Type Culture Collection (ATCC).

### 2.2. Oligonucleotides Transfection

The day before transfection, the cells were seeded and then transfected according to the manufacturer’s instruction (JetPrime Polyplus transfection protocol, Strasbourg, France). Briefly, oligonucleotides were diluted into 200 µL of jetPrime Buffer (supplied) and mixed by vortex. The JetPrime transfection complex was prepared by adding 4 μL of JetPrime transfection reagent into the mix containing JetPrime buffer and oligonucleotides and further incubated for 10 min at Room Temperature (RT). After incubation, JetPrime transfection complex was added into the medium and distributed evenly. The final concentration of oligonucleotides used for transfection was 200 nM. The cells were incubated at 37 °C and 5% CO_2_ until the analysis was carried out (24 and 48 h after transfection) without the need to change the transfection medium. Oligonucleotide sequences are available on request.

### 2.3. OXA, and PTX Treatment in Presence or Not of Oligonucleotide Antisense

Cells were seeded 18 h before chemotherapy drug treatment in an appropriate concentration. OXA was used in a range of 0.1 to 25 µM in all cell lines. PTX treatments are between 0.01 and 0.1 µM. At the end of 24 h treatment, the IC_50_ value was calculated using GraphPad V.8 program (San Diego, CA, USA). In combined treatment, OXA and PTX were added after the medium change at 20 h for an additional 24 h and 48 h.

### 2.4. Protein Expression Analysis by Western Blot 

Proteins from the different cell lines were fractionated by SDS-PAGE and transferred to Nitrocellulose membrane (Biorad Laboratories, Hercules, CA, USA), according to Botta et al. 2019 [26]. After blotting, the membrane was incubated with appropriate primary and secondary antibodies. Primary antibodies used were against CSA (Cell signaling Techology, Danvers, MA, USA) and β-actin (Santa Cruz Biotecnology, Santa Cruz, CA, USA), while secondary antibodies were HRP conjugated.

### 2.5. Retrotranscription and Real-Time Quantitative PCR

RNA was isolated using NucleoSpin RNA kit (Macherey-Nagel, GmBH & CO., Dueren, Germany) according to the manufacturer’s instructions. RNA was diluted by adding 40 µL of DEPC water and its integrity was checked on a denaturing 1% agarose gel. RNA concentration was measured with Qubit Fluorometer 2.0 cDNA synthesis, which was performed with 1 µg of RNA for each sample by using the First Strand cDNA Synthesis kit (Thermo Fisher Scientific, Waltham, MA, USA). Comparative qRT-PCR was carried out with GoTaq qPCR SYBR green master mix (Promega, Madison, WI, USA), using Mx3005P Real-Time PCR system (Agilent, Santa Clara, CA, USA). Results were normalized to *β-Actin*. Primer sequences are available on request.

### 2.6. Cell Viability MTT Assay

Cell viability was evaluated by measuring the mitochondrial-dependent conversion of the yellow tetrazolium salt MTT [3-(4.5-dimethylthiazol-2-yl)-2.5-diphenyl-2H-tetrazolium bromide] to a purple formazan crystal by metabolically active cells. The procedure was performed according to Caputo et al. [27]. The experiments were conducted in triplicate.

### 2.7. Proliferation Tripan Blue Exclusion (TBE) Assay

The day before transfection, 1 × 10^5^ cells were plated in 6-well dishes. The cell proliferation assay was performed using 0.05% Trypan Blue solution to distinguish live and dead cells within a Burker chamber. Around 500 cells for three replicate counts were determined at each time point (24 h and 48 h after transfection).

### 2.8. Apoptosis Assay

A combination of Fluorescein Diacetate (FDA; 15 μg/mL), Propidium Iodide (PI, 5 μg/mL), and Hoechst (HO, 2 μg/mL) was used to differentiate apoptotic cells from viable cells. FDA and HO are vital dyes that stain the cytoplasm and nucleus of the viable cells, respectively. The necrotic and the late stage of apoptotic cells are readily identified by PI staining. Approximately 500 randomly chosen cells were microscopically analyzed to determine apoptosis levels. The procedure was performed according to Proietti-De-Santis et al. [28].

### 2.9. Statistical Analysis

Statistical analysis was performed by using the SPSS package (IBM-SPSS Inc., Armonk, NY, USA). Shapiro–Wilk test was firstly used to verify the normality of the data.

In order to verify the presence between MCF10A and the cancer cell lines on the CSA protein relative quantification, CSA mRNA relative quantification, and relative cell viability in cases of OXA and PTX conditions, *t*-tests were performed. Even if it is true that multiple comparisons should be added, we only seek to understand the presence of difference between MCF10A and each one of the cancer cell lines, independently. Successively, a two-way ANOVA was conducted in order to verify the presence of statistical difference on the relative cell viability considering the cell lines and the treatment as main effects. Specifically, the cell lines considered four (MCF10A, MCF-7, MDA-MB231, and T47D) and three levels (no treatment, SO and ASO) for cell lines and treatment, respectively. In case of significant interaction effects, two one-way ANOVA tests were performed, independently for each main effect. A multiple comparison Bonferroni test was applied when ANOVA was significant. The same tests were also performed when comparing the effects of the four cell lines and the treatment (medium, 0.01 µM, 0.01 µM + ASO) in OXA and PTX conditions on relative cell viability.

All the statistical tests were conducted by setting the significance level at 0.05.

## 3. Results

### 3.1. CSA Gene Is Overexpressed in Breast Cancer Cells

In order to assess whether CSA was overexpressed in BC cells, we measured CSA protein levels by Western blotting. We compared BC cell lines MCF-7 (ER+, PR- and Her2-), MDA-MB231 (ER-, PR-and Her2-), and T47D (ER+/PR+ and Her2-) with normal breast MCF-10A cell line. As shown in Figure 1, Western blot analysis using total cellular extracts (Figure 1A) and relative protein quantification (Figure 1B) revealed an increase of CSA protein levels in all the BC cell lines analyzed of 2.01 ± 0.15, 4.18 ± 0.76, and 5.39 ± 0.3 for MCF-7, MDA-MB231, and T47D, respectively, when compared to MCF-10A. To confirm that CSA over-expression was correlated to an increase of mRNA levels, a qRT-PCR analysis was performed. All BC cell lines analyzed displayed an overall increase of *CSA* mRNA levels in a range of 1.41 ± 0.12, 3.8 ± 0.42 and 4.2 ± 1.5 in MCF-7, MDA-MB231, and T47D cell lines respectively, when compared to MCF-10A (Figure 1C).

### 3.2. CSA Silencing Reduced Viability and Proliferation in BC Cell Lines

Being aware that CSA protein negatively modulates pro-apoptotic signaling, we wondered whether CSA overexpression was responsible for the increased robustness exhibited by BC cell lines. We decided to suppress CSA expression and test if the absence of CSA would reduce the tumorigenicity of BC cells by tipping the balance towards cell cycle arrest and death and away from cell proliferation and survival. To inhibit CSA expression, we designed an antisense oligonucleotide (ASO) to target and induce the degradation of its mRNA. An oligonucleotide matching the sense sequence (SO) of *CSA* mRNA was used as control. Figure 2A shows *CSA* mRNA expression levels in all the cell lines analyzed 24 h later from transfection with either ASO or SO (final concentration 200 nM). As shown, ASO treatment efficiently reduced *CSA* mRNA levels ranging from 40 to 60% in all cell lines analyzed.

Subsequently, we studied the effects of CSA ablation on cell viability and proliferation in both breast normal and cancer cell lines. Cell viability was analyzed by MTT assay at 24 h post-oligonucleotides transfection. Results shown in Figure 2B displayed a dramatic decrease of cancer cell survival, of 55.5% ± 7.2, 66.8% ± 6, and 66.9% ± 6.2 for MCF-7, MDA-MB231, and T47D cell lines, respectively, as compared to the mock-treated ones, while there was no effect on normal breast MCF-10A cell line. In contrast, SO treatment, used as a negative control to ascertain that the results were due to the specific inhibition of CSA and not to the transfection procedure, had no effect in terms of cell viability in all the cell lines analyzed. These results are confirmed by the statistical analysis. In fact, the interaction factor was found significant (*p* < 0.01) and the ANOVA analyses revealed statistical differences due to the ASO treatment on the relative viability (*p* < 0.01) with respect to no treatment and the SO. Focusing on the line cells, only differences between MCF10A and the three breast cancers (MCF-7, MDA-MB231 and T47D) were found (*p*-value ranged from 0.001 to 0.01). In addition, inhibition of proliferation in normal and BC cells was measured by directly counting viable cells at different times. Briefly, inhibition of CSA protein expression determined a decrease of cellular proliferation in MCF-7, MDA-MB231, and T47D BC cell lines, but not in normal breast MCF-10A cell line, at either 24 h and 48 h after ASO transfection, while the transfection of the respective SO had no effect (Figure 3).

### 3.3. CSA Antisense Targeting Enhances Anticancer Drug Sensitivity in Breast Cancer Cell Lines

To investigate whether down-regulation of CSA expression has the potential to selectively sensitize BC cells to low doses of chemotherapy, which are not harmful for normal breast cells, a combined treatment with ASO and low doses of different chemotherapy drugs, paclitaxel (PTX) or oxaliplatin (OXA), was performed.

Initially, we performed single treatment with increasing doses of either OXA or PTX in normal breast MCF-10A cell line to select a dose that is not harmful. The dose response curves for each chemotherapy drug tested, obtained by MTT assay, are shown in Figure 4 (panels A and B). The MCF-10A cell viability decrease shows a behavior dose-dependent for both the chemotherapeutic drugs. Based on these results, we decided to choose the lowest dose for each drug: 0.01 µM for PTX and 2.5 µM for OXA. So to investigate if down-regulation of CSA expression does sensitize MCF-7, MDA-MB231, and T47D BC cells to chemotherapy, a combined treatment with ASO and each of the chemotherapy drugs above-mentioned was performed.

Figure 4C shows cell viability after ASO and OXA combined treatment. Interestingly, CSA ablation effectively sensitized BC cells to low doses (2.5 µM) of OXA, thus reducing cell viability if compared to the drug treatment alone.

Regarding the combined (ASO and OXA) treatment, the decrease was from 96.5 vs. 53.3% in MCF7 cells, from 93.6 vs. 52.2% in MDA-MB231, and from 98.4 vs. 55.3% in T47D cells. Noteworthy, ASO treatment did not further sensitize MCF-10A to OXA treatment. The same graph also illustrates that SO treatment did not sensitize BC cells to OXA. These results are confirmed by the statistical analysis. In fact, the interaction factor was found significant (*p* < 0.01) and the ANOVA analyses revealed statistical differences among all the treatments on the relative viability (*p* < 0.01). Focusing on the line cells, only differences between MCF10A and the three breast cancers (MCF-7, MDA-MB231, and T47D) were found (*p*-value always lower than 0.001).

In Figure 4D, cell survival after ASO and PTX combined treatment is shown. Also in this case, CSA ablation effectively sensitized BC cells to low doses (0.01 µM) of PTX, dramatically decreasing cell viability, if compared to the single drug alone. Regarding the ASO and PTX combined treatment, the reduction was from 70.05 vs. 50.3% in MCF7 cells, from 91.5 vs. 58.8% in MDA-MB231, and from 82.5 vs. 61% in T47D cells. Again, ASO treatment did not further sensitize MCF-10A cell line to PTX. The statistical results arising from ANOVA are similar to the ones reported for the combined treatment ASO and OXA.

Ideally, a chemotherapy drug should only kill the cancer cells, without affecting normal cells. For this purpose, Selectivity Index (SI), defined as the ratio of the toxic concentration of a compound to its effective bioactive concentration, was calculated by comparing chemotherapy drug half-inhibitor concentration (IC_50_) value in the normal cell line against the IC_50_ of the same compound in the cancer cell line. The ideal drug should have a relatively high toxic concentration with a very low active concentration. In this regard, Weerapreeyakul et al. [29] proposed that only a drug with a SI value equal to or greater than 3.0 should be considered worthy of further investigation. To assess the selectivity (cancer vs. normal breast cells) of CSA ablation in enhancing the anticancer activity of conventional chemotherapy drugs, their cytotoxicity was determined in both normal and BC cell lines either combined or not with the ASO treatment. With this aim, we calculated the IC_50_ of both OXA (Table 1A) and PTX (Table 1B) chemotherapy drugs in all the cell lines. As shown, the ASO treatment overall decreased the IC_50_ in all BC cell lines examined. Interestingly, CSA silencing dramatically sensitized BC cells to OXA by increasing SI from 0.74 to 4.99 in MCF7, from 1.92 to 5.4 in MDA-MB231, and from 0.62 to 5.5 in T47D. Regarding PTX, ASO treatment dramatically increased the sensitivity and selectivity in MDA-MB231 and T47D, shifting the SI from 2.5 to 8.0 and from 3.4 to 8.0, respectively, whereas, in the case of MCF-7, the increased sensibility and selectivity is only marginal (from 6.4 to 8), because this cell line yet displayed a good sensitivity to PTX treatment. With the aim to analyze if CSA inhibition sensitizes chemo-resistant cells to chemotherapy treatment, we developed a cellular model resistant to OXA by repeatedly exposing MDA-MB231 cells to increasing drug concentrations. IC_50_ value (54.89 µM) of the surviving daughter-resistant cells (MDA-MB231^resistant^) showed a 2.5-fold increase in resistance to OXA when compared to the parental sensitive cells (IC_50_ = 21.91 µM). Interestingly, CSA ablation by ASO treatment restored the sensitivity of MDA-MB231^resistant^ cells to a level similar to the one displayed by normal MDA-MB231 cells (24.06 vs. 21.91 µM) (Table 1C).

### 3.4. CSA Suppression Massively Induces Apoptosis in BC Cells When Treated with PTX and OXA

Since CSA protein plays a relevant role in the negative regulation of apoptosis, we wanted to investigate what happens in response to chemotherapy when CSA expression is suppressed.

As shown in Figure 5 (panels A and B), CSA silencing dramatically increased the levels of cleaved Caspase-3, an unequivocal marker of apoptosis, displayed by MDA-MB231 cells after PTX and OXA treatment. In addition, we also quantitatively assessed the rates of apoptosis by using a combination of fluorescent dyes to analyze the morphological alterations that cells undergo during apoptosis: FDA and HO stain the cytoplasm and the nucleus, respectively, of viable cells, while the necrotic and the late-stage apoptotic cells are stained by PI, which requires the loss of cytoplasm and nuclear membrane integrity. Our results show that, compared to the mock-treated cells (Figure 5C, panel a), the yet largely significant increase of the apoptosis rate in PTX and OXA-treated cells (Figure 5C, panels c and e) further increases in the case of ASO co-treatment (Figure 5C, panels d and f). Results of the fluorescent assay are graphed in Figure 5D. Altogether, these data show that CSA suppression dramatically enhances the apoptosis triggering in BC cells treated with PTX and OXA.

## 4. Discussion

In this paper, we showed that CSA, the complementation group that, together with CSB, is associated to CS, is up-regulated in BC. We demonstrated that CSA ablation *per se* reduces the tumorigenicity of cancerous cells by dramatically affecting their survival and proliferation. Therefore, as previously demonstrated for CSB, also CSA is upregulated in cancer cells and its ablation seems to perturbate their robustness.

The question that arises is why should tumors benefit from CSA over-functionality? Interestingly, CSA is a member of the mixed class of intrinsically disordered proteins (IDPs) as it contains intrinsically disordered internal segments and a long floating tail along with ordered regions. IDPs are characterized by a remarkable conformational flexibility and structural plasticity, which allows them to have a huge number of molecular partners and to integrate different biological functions. This is why IDPs play critical roles in regulation and signaling [30]. The integration of various biological processes is a distinctive feature of cancer genes and the important role played by IDPs in this regard is becoming increasingly clear. Interestingly, disordered regions have been strongly associated with dosage-sensitivity oncogenes in cancer [31,32]. It is therefore possible that dosage sensitivity is related to the ability of proteins to perform molecular interactions via linear sequence motifs as their concentration increases. In this context, we and others have already showed that CSA and CSB proteins play multiple roles in modulating cell robustness through the positive regulation of pro-survival adaptive responses including DNA repair, hypoxia response, UPR activation, and negative regulation of pro-apoptotic pathways, such as the ones triggered by p53 and ATF3 [17,19]. All these characteristics presumably make cancer cells addicted to CS proteins over-functionality. Along this line, the over-expression of CSA in BC cells could be the result of the selective pressure that tumor cells undergo to overcome the adverse conditions met during cancer progression.

Moreover, another interesting finding of this work is that CSA ablation sensitizes BC cells to the chemotherapy attack inferred with OXA and PTX drugs, regardless of their genetic background. We hypothesize two main mechanisms underlying the sensitization to OXA after CSA suppression. First, given that TC-NER pathway, which depends on the functionality of CS proteins, is one of the major mechanisms involved in the repair of OXA-induced DNA adducts, CSA ablation might result in a higher cytotoxic effect of this drug. Accordingly, Slikova and co-workers [33] recently demonstrated that CSA protein plays a protective role against OXA in colon adenocarcinoma cells.

Second, given the role of CSA in the negative regulation of apoptosis [18], its suppression might reduce the anti-apoptotic buffering capacity of the cells, making them more susceptible to cytotoxic insults. Ablation of CSA drastically sensitizes BC cells also to PTX. Certainly, the decrease of the anti-apoptotic buffering capacity correlated with CSA ablation should be one of the mechanisms underlying the potentiation of PTX cytotoxicity. The potentiation of PTX conferred by CSA ablation could be also correlated to the recent role discovered for CS proteins in promoting cytokinesis by determining the cleavage of the intercellular bridge between the two daughter cells. Since also PTX, by hampering microtubules depolymerization, interferes with intercellular bridge dynamics, we can conclude that CSA ablation and PTX treatment would additively impair cell division.

Another interesting aspect of our findings is that ablation of CSA does enhance OXA and PTX selectivity for cancer cells, as shown in Table 1. Indeed, an overall decrease of the IC_50_ was observed in all BC cell lines, but not in the normal cell line, when the drug was inferred in combination with CSA ASO. The selective killing of cancer cells without damaging normal cells is the most important feature for cancer treatment, decreasing the onset of chemoresistence.

## 5. Conclusions

Overall, these findings suggest that CSA ablation in BC cells could improve the magnitude of therapeutic responses and reduce the likelihood of acquired resistance in an individual patient. The fact that CSA ablation seems to sensitize BC cells regardless of their genetic background is particularly interesting, given the genetic inter- and intra-tumor heterogeneity exhibited by BC. This aspect has a strong relevance in the case of Triple-negative subtype of BC, being that, due to the negative nature of the three major receptors (ER, PR and HER2), its treatment stands mainly on the conventional chemotherapy. Based on these findings, we can conclude that CSA may be a very attractive target for the development of new specific anticancer therapies that combine the use of antisense oligonucleotide or specific inhibitors of CSA function with chemotherapy drugs, with the aim to make the treatment for BC more effective. Certainly, the safety issue arises some concerns and needs to be deepened. Ablation of CSA does not appear to affect normal epithelial cells. However, being that CSA is virtually expressed in all tissues, it is mandatory to assess in vivo whether this therapeutic approach could also hit quiescent and slow proliferating cells that are usually resistant to conventional chemotherapy drugs. 

## Figures and Tables

**Figure 1 cancers-14-01687-f001:**
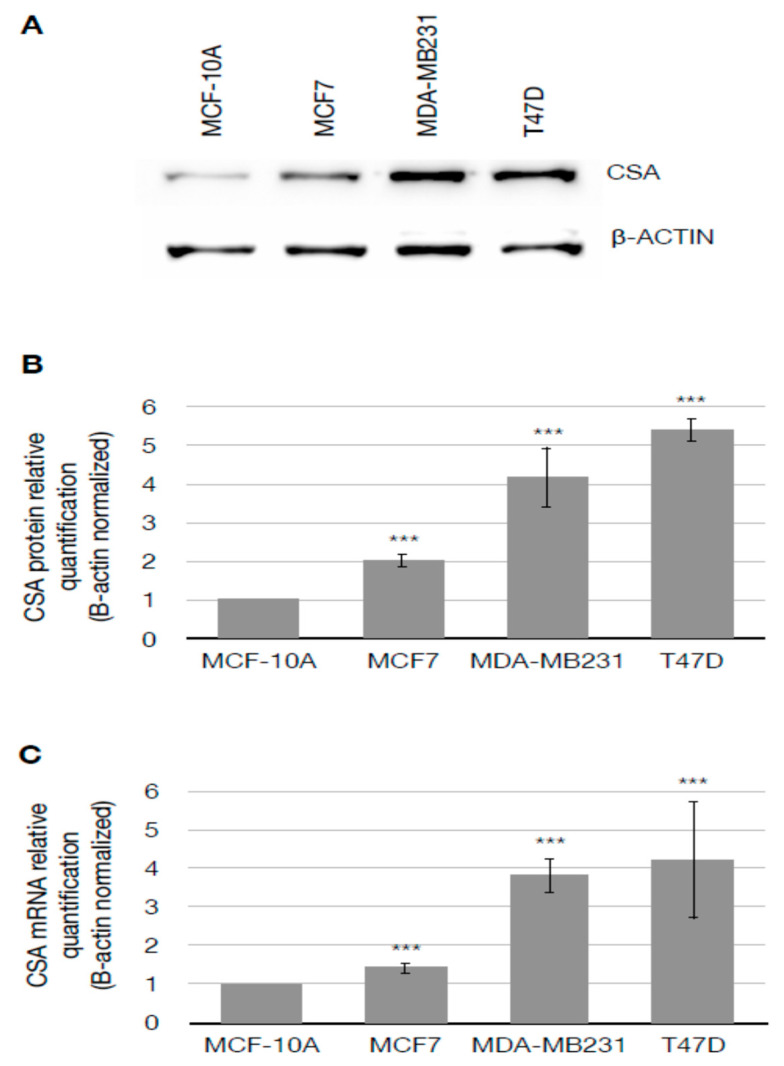
(**A**) Western Blotting showing CSA protein and β-Actin levels. β-Actin was used as loading control; Original blots see Appendix A. (**B**) Graph showing CSA relative amount β-Actin normalized; (**C**) qRT-PCR analysis of *CSA* mRNA expression. Data are presented as means ± SD of three independent experiments. *** is for *p* < 0.001 calculated with Student’s T Test.

**Figure 2 cancers-14-01687-f002:**
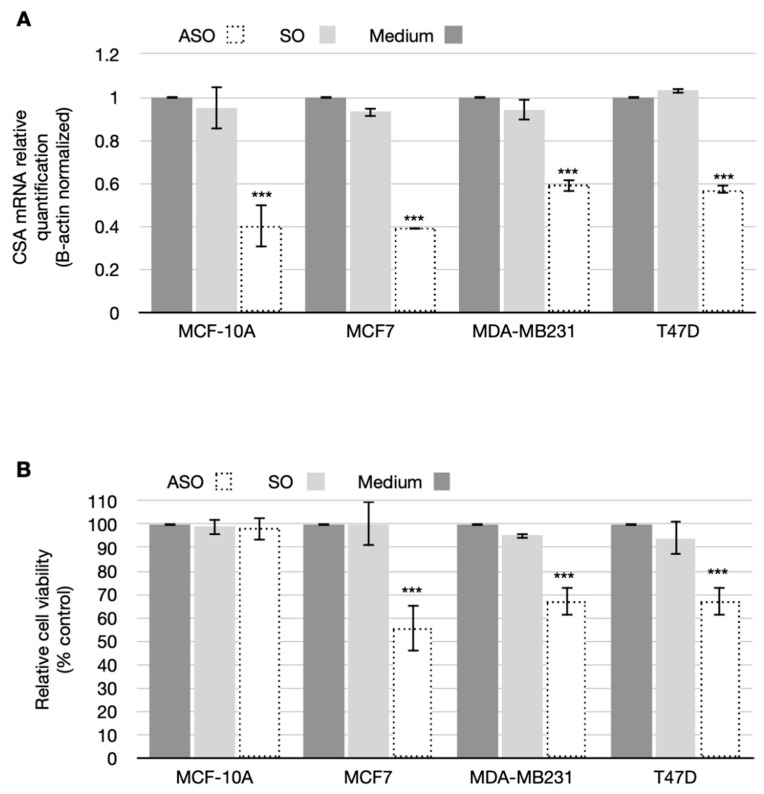
(**A**) qRT-PCR analysis of *CSA* mRNA expression 24 h after transfection. The results, *β-Actin* normalized, are presented as means (± SD) of three independent experiments. *** is for *p* < 0.001 calculated with ANOVA test. (**B**) Graph showing relative cell viability obtained by MTT assay after 24 h of transfection. Data are presented as means ± SD of three independent experiments. *** is for *p* < 0.001 calculated with ANOVA tests.

**Figure 3 cancers-14-01687-f003:**
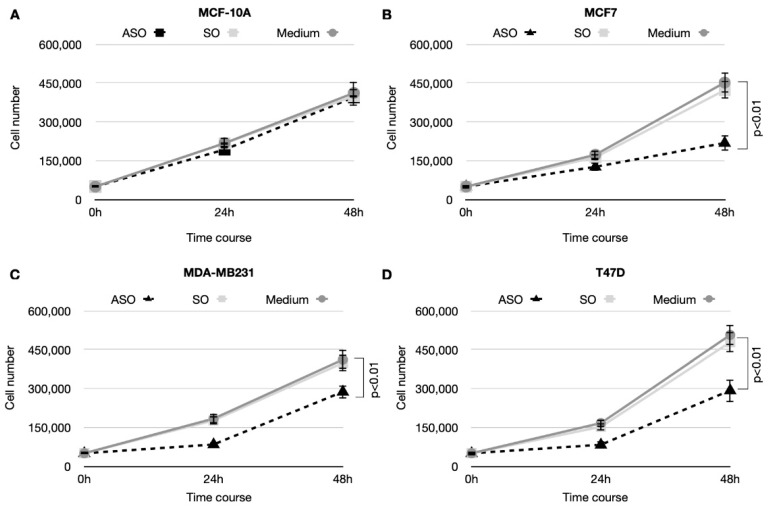
Graphs showing cell proliferation assay in (**A**) non-tumorigenic cells (MCF-10A) and in (**B**–**D**) breast cancer cell lines (MCF7, MDA-MB231, and T47D). Cells were counted at the time of seeding and after 24 and 48 h after transfection. Data are presented as means ± SD of three independent experiments, calculated with χ^2^ test.

**Figure 4 cancers-14-01687-f004:**
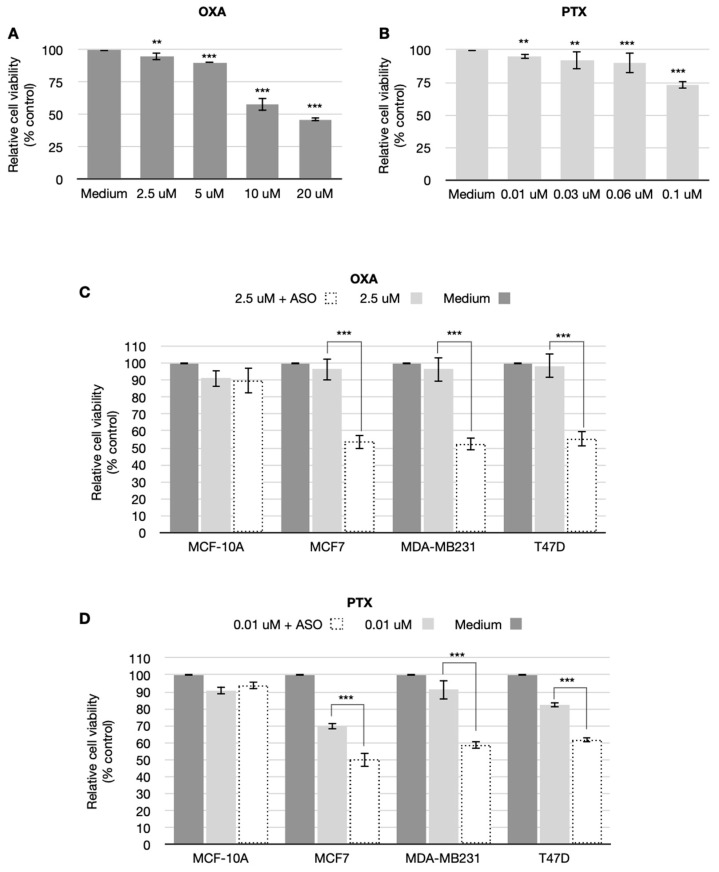
(**A**,**B**) Graph showing relative cell viability obtained by MTT assay in MCF-10A cell line after 24 h of either OXA or PTX treatment. Data are presented as means ± SD of three independent experiments. ** is for *p* < 0.01, *** is for *p* < 0.001 calculated with χ^2^ test. (**C**,**D**) Graph showing relative cell viability obtained by MTT assay in MCF-10A and MCF-7, MDA-MB231, and T-47D cell lines after 24 h of either OXA or PTX alone and combined CSA ASO plus OXA or PTX treatments. Data are presented as means ± SD of three independent experiments. *** is for *p* < 0.001 calculated with ANOVA tests.

**Figure 5 cancers-14-01687-f005:**
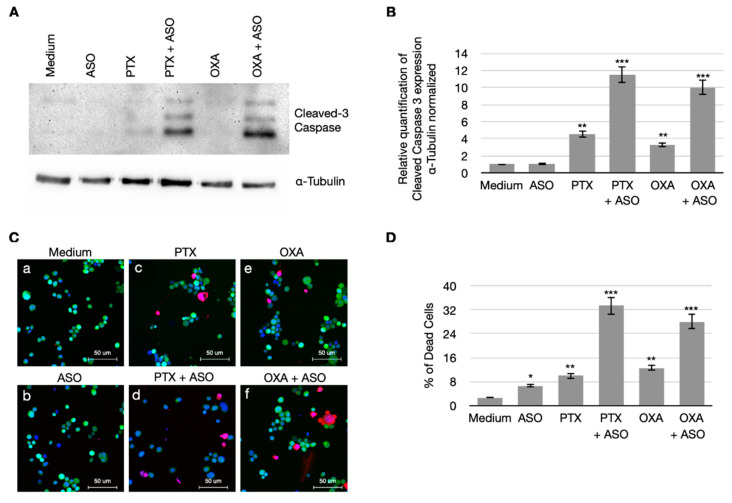
CSA ablation determines a massive apoptosis induction after ASO with PTX or OXA combined treatment in Triple Negative Breast cancer cells (MDA-MB231). Cleaved caspase 3 protein expression levels in MDA-MB231 total cellular extracts after single or combined treatment. Western Blotting was performed using antibodies against cleaved 3-Caspase (Cell Signaling) and α-Tubulin (Sigma Aldrich, St. Louis, MO, USA). The results are the representation of three independent biological repeats (**A**). Graph showing levels of cleaved caspase 3, α-tubulin normalized, in MDA-MB231 cells after treatment. Original blots see Appendix A. (**B**). Confocal micrographs of MDA-MB231 cells after treatment, stained for apoptosis assay with Hoechest (HO, blue), Fluorescein diacetate (FDA, green) and Propidium Iodide (PI, red) (**C**). Graph showing the percentage of cell death from apoptosis assay (n = 500 × 3 independent experiments) (**D**). N.S: not significant, * *p* < 0.05, ** *p* < 0.01, *** *p* < 0.001.

**Table 1 cancers-14-01687-t001:** (**A**–**C**) Tables of IC_50_ ± SD (half maximal inhibitory concentration ± SD) values for all the compounds were expressed in μM unit. SI (Selectivity Index) was calculated as a ratio of IC_50_ (non-tumorigenic cell line)/IC_50_ (cancer cell line). In bold were SI < 3, indicating that the drug is not cancer selective.

**(A) OXA**		
**Cell Line**	**IC_50_ (** **µ** **M)**	**SI**
MCF-10A	42.10 ± 0.66	
MCF-10A + ASO	19.99 ± 1.0	
MCF-7	56.85 ± 1.4	**0.74**
MCF7 + ASO	4 ± 0.04	4.99
MDA-MB231	21.91 ± 0.46	**1.92**
MDA-MB231 + ASO	3.7 ± 0.29	5.4
T47D	67.35 ± 1.04	**0.62**
T47D + ASO	3.6 ± 0.02	5.5
**(B) PTX**		
**Cell Line**	**IC_50_ (** **µ** **M)**	**SI**
MCF-10A	0.51 ± 0.04	
MCF-10A + ASO	0.16 ± 0.05	
MCF-7	0.08 ± 0.02	6.4
MCF7 + ASO	0.02 ± 0.001	8
MDA-MB231	0.20 ± 0.04	**2.55**
MDA-MB231 + ASO	0.02 ± 0.002	8
T47D	0.15 ± 0.02	3.4
T47D + ASO	0.02 ± 0.003	8
**(C) OXA**		
**Cell Line**	**IC_50_ (** **µ** **M)**	
MDA-MB321	21.91 ± 0.46	
MDA-MB231^resistant^	54.9 ± 0.56	
MDA-MB231^resistant^ + ASO	22.90 ± 0.90	

## Data Availability

The data presented in this study are available on request from the corresponding author.

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
