# Peer review of "CSA Antisense Targeting Enhances Anticancer Drug Sensitivity in Breast Cancer Cells, including the Triple-Negative Subtype"

_cancers, 2022, doi:10.3390/cancers14071687_

Round 1

Reviewer 1 Report

I thank the authors for their consideration of my suggestions. The discussion is now clearer and opens to a future direction to investigate. I think the manuscript is now publishable.

Author Response

Thanks for your review.

Reviewer 2 Report

Minor concerns:

- missing ASO "single-treated" as a control in all panels of figure 5.

Author Response

We replaces Fig.5 as required

This manuscript is a resubmission of an earlier submission. The following is a list of the peer review reports and author responses from that submission.

Round 1

Reviewer 1 Report

Silvia et al reported the expression of Cockayne Syndrome group A (CSA) in several breast cancer cell lines and found that knockdown of CSA protein by ASO in these BC cell lines affected the proliferation and that ASO appeared to cooperate with existing chemotherapeutic drugs such as OXA and PTX. The authors proposed that CSA may be a novel target for treating breast cancer. I found this study is relatively superficial that it failed to provide any molecular insights on how CSA was affecting BC cell survival. Furthermore, without supporting data from any in vivo tumor model, it would be hard to conclude that such strategy may be clinical relevant to BC treatment. Hence, I do not support its publication in Cancers given the study are still too preliminary at this stage. 

Minor comments: the English language must be improved since there are plenty of grammatical mistakes throughout the paper. 

For example: 1. In Abstract line 14: " This issue results particularly relevant in....", I think the authors are trying to say" This issue is particularly relevant in....

2. Keyword: "oligonucleotide anti-sense" should be " anti-sense oligonucleotide (ASO)"

3. In Many Figure legends, "t-Student test" should be" Student's T Test". 

4. Pls add title for all Figure legends. 

5. Figure 1: pls show all original western blot results for the quantification shown in B.  

Reviewer 2 Report

The authors use advanced drug technology to find a solution to chemoresistance in TNBC. The experimental design is clear, the experiments are elegant and well designed. Interestingly, all cultured BC models have CSA over-expressed. The results are important to the clinic, but it's about turning them into something usable. I realize that the authors have used some technological skills already possessed, applying them to a new field. As I also understand that with ASO technology in cancer, we are still at the beginning. The result of the combined action of OSA + chemo with a change of about -50% is very interesting; as well as that ASO treatment did not further sensitize MCF-10A to either drug. Finally, SO treatment did not sensitize BC cells to OXA.

Two important points:

First point - The discussion is clear and smooth but the Authors should improve it. The anti-apoptotic buffering capacity correlates with CSA ablation and the authors suggest various possible functional mechanisms. The functional implications reflect the structural properties of the complexes and of the individual proteins that actuate them. Authors are opening an alternative path to fight TNBC. But we still need more information to complete the picture. I did a quick interactome analysis on STRING (I have attached the file about UniProt. Q13216). It shows that CSA seems involved in very few cancers (perhaps because it was little studied in cancer), but ERCC8 is also a key protein in many other molecular metabolic mechanisms. When a new possibility opens up, Authors should dare by going a little further with some additional explanation and/or hypothesis. Now a day, we know the particular structural characteristics of ERCC8. It is a member of the mixed-disordered protein class; thus, we have elements to explain the protein over-functionality and try to clarify its great functional versatility. This cannot be omitted from the discussion.

2nd point - A scientific article is aimed at the scientific community that asks itself questions during its reading. They certainly aim to understand what is after. Targeting mRNA increases the number and type of treatable diseases ("regardless of their genetic background"), but we don't know the individual phenotypic response in vivo. This is the real limit, in particular for TNBC. ASO treatment does not appear to affect normal MCF-10 epithelial cells, but ERCC8 expression is virtually omnipresent in all tissues. Do the authors have an explanation? What predictions we can make? How do the authors think we can use such a type of drug in vivo? At what concentrations do they think it can have an appreciable effect? (The in vitro effect is 40 - 60%).

Therefore, I suggest reviewing the Discussion by trying to answer and explain these aspects. The pdf file I am also attaching contains a rough functional check carried out on STRING through PPI networking. The file illustrates the many functional roles of ERCC8, more than what we can realize. This motivates my requests.

Reviewer 3 Report

In the work of Filippi and colleagues, the potential targeting of CSA was assessed in breast cancer cell lines. The manuscript is well-written and straightforward. The conclusions are supported by the author's findings but I have a few suggestions to improve the robustness of the paper, before proceeding with publication.

  • Since CSA has a relevant role in negatively regulating apoptosis and the authors speculated the importance of this aspect in the chemotherapy response, they should check and show the apoptosis levels in CSA knock-down cells (i.e. AnnexinV staining or caspase activation).
  • Try to avoid saying that chemotherapy is the “only” therapeutic option for TNBC patients. It is more correct to say “main” or “principal”. Please correct it throughout the manuscript.

Reviewer 4 Report

In this manuscript, the authors showed, through in vitro experiments, that breast cancer cell lines overexpress Cockayne Syndrome Group A protein (CSA). Inhibition of CSA protein expression sensitizes breast cancer cells to platinum and taxane treatment. There are some major issues in the presentation of the data which negatively affected the quality of the study.

  1. the quality of the figures are too low;
  2. the statistical analysis is not appropriate. ANOVA with post hoc needs to be used for multiple comparison;
  3. the paclitaxel concentration is too low to generate any meaningful IC50 values.
  4. at least one chemo-resistant BC cell line need to be included to test the sensitizing effect of CSA inhibition.